# From STEAM to CHEER: A Case Study of Design Education Development in Taiwan

Chinlon Lin, Jianping Huang and Rungtai Lin *

Graduate School of Creative Industry Design, National Taiwan University of Arts,
New Taipei City 22058, Taiwan; 10735903@mail.ntua.edu.tw (C.L.); 10735905@mail.ntua.edu.tw (J.H.)
* Correspondence: rtlin@mail.ntua.edu.tw

**Abstract:** The purpose of this study is to provide other developing countries in the region a reference on the successful design education reformation in Taiwan. The study first reviews Taiwan's economic and design development to show their interconnection with local culture and the global market. Next, the study explores Taiwan's design education development which transforms from adopting STEM (science, technology, engineering, mathematics) to STEAM (science, technology, engineering, art, mathematics) in secondary school, and understands how it overcomes challenges by the help of both public and private sectors. Last, a conceptual framework is proposed to present and study Taiwan's design education development. The result from case studies validates that STEAM can reach SAD (science, arts, and design) in higher education and CHEER (collaboration, humanity, empathy, ecology, and renaissance) in design practice. Therefore, the result and implications provided by this study can serve as a reference for other countries in the region who share similar cultural and socioeconomic development and future goals as Taiwan.

**Keywords:** STEM; STEAM; design education; gender equality; interdisciplinary

## 1. Introduction

Taiwan is a multi-cultural blend of Chinese with significant East Asian influences. Its growing multiculturalism is shown not only in politics, but also in its socio-cultural aspects. Although political uncertainty has always been a challenge for Taiwan due to its diplomatic isolation, the people have shown solidarity through action [1]. For example, a recent advertisement in The New York Times entitled 'Who can help? Taiwan', along with its slogan 'Taiwan Can Help', was made possible through a crowdfunding campaign initiated by a young Taiwanese designer. This event shows that the younger generation can react with innovative ideas when facing challenges [2]. In recent years, Taiwan's design teams have been the frequent winners of international design awards, and designers' competency is witnessed and proven in the global design arena through contests [3]. These achievements are great encouragement for the younger generation and living proof of Taiwan's successful design education system.

Having a successful education system is the key factor to the prosperity of any developing country's economy. According to the Southeast Asian Ministers of Education Organization's (SEAMEO) New Education Agenda [4], strengthening countries from the Association of Southeast Asian Nations (ASEAN)'s technical and vocational education and training (TVET) to meet the demand of Industry Revolution 4.0 is the most prevailing issue and the priority task to move the region's economy further and faster. Ms. Lorna Dig-Dino, Undersecretary of the Philippines' Department of Education, stated that graduates of secondary school should be ready for higher education and employment in terms of their 21st century skills, which includes not only technical skills, but also emotional soft skills which are timeless skills and required by business and industry [4]. Chamnarong Pornrungroj, Director of Office for National Education Standards and Quality Assessment (ONESQA) in Thailand, pointed out that shortage in the labor market has a serious impact

on the country's competency; if Thailand's TVET cannot produce enough competitive workforce to meet the market's demand, the country would have to seek help from foreign countries [5]. These show the significance of a functional education system to the economy of developing countries.

Science and technology are two elements that shape the future of society and economic growth. Increasing women's enrollment in STEM (science, technology, engineering, and mathematic) can lead to better solutions to problems, since the potential for innovation is larger [6]. However, research from United Nation's Educational, Scientific, and Cultural Organization (UNESCO) shows that female students represent only 35% of all students enrolled in STEM-related degrees worldwide. Lee [7] points out that if women's participation in these disciplines fell behind the speed of economic growth, the gender gap and workforce shortage would only become larger, which would hinder economic development. Lack of female professionals also reduces the diversity of perspectives and the ability to offer different answers and breadth to new problems [6]. Kersey et al. [8] indicated that boys and girls do not differ in early quantitative and mathematical ability, and in fact they are equally equipped to reason about mathematics during early childhood. Gender disparities in the interest towards STEM disciplines are substantial [7]. In response to this issue, Taiwan introduced STEAM (science, technology, engineering, arts, and mathematic) to the secondary level education, and placed great emphasis on arts and design-related disciplines.

Taiwan's 'Economic Miracle', which was driven by new innovation in science and technologies that create new experiences and value for the global market, was only possible because of the people's hard work and the country's smooth transition in the education system, higher education in particular. Taiwan placed great emphasis on industrial production in the past, but the trend shifted when design became an important source of national competency [9]. Currently, design-related disciplines are flourishing in Taiwan's higher education. There are a number of well-established design educational programs available in universities, ranging from undergraduate, postgraduate, and even doctoral degrees [10]. Now is the time for Taiwan's achievements in the design education system to be recognized globally and to serve as a model for other developing countries such as the ASEAN who are seeking to improve their long-term economic growth and prosperity through educational reformation.

Based on Taiwan's historical development and other previous studies [2,3,11–13], this research intends to first explore how Taiwan's economy advanced from 'function' to 'feeling' and how gender issue with STEM was tackled, which both consequently promoted the evolution of its design education. Next, case studies are applied to validate Taiwan's design development as a fusion of 'function' and 'feeling'. The result shows that the interwoven relationship between 'globalization' and 'localization' in Taiwan's design education system is the key factor to the success of Taiwan's economy, and the framework constructed can serve as a model in educational reformation for the global community.

## 2. Background—Taiwan's Economic and Design Development

Taiwan's "Economic Miracle", a rapid industrialization and economic growth with major market structural change during the mid-1950s to the mid-1980s, is regarded as a typical pattern for a developing country [14,15]. Schive [16] argued that market-friendly and self-restrained industry policies were the key factors to the success of Taiwan's fast industrialization. Chen et al. [17] conducted a project entitled 'Ideastorming: Concept Design of Future Products' which was funded by the National Science Council of Taiwan to foster interdisciplinary and collaborative innovation between design and technology on campus. In 2011, a project called 'Dechnology' (design + technology) was supported by the Department of Industrial Technology under the Ministry of Economic Affairs (MOEA), and executed by the Industrial Technology Research Institute (ITRI) [18]. Kreifeldt et al. [12] proposed an approach to build an interdisciplinary relationship between 'technology' in design education and 'human-centered' in design practice. For 'technology' to be

realized in education, design thinking must be applied to design problem-solving while the 'human-centered' concept is used to cultivate the collaboration between 'humanity' and 'art' (humart) in design practice. This integration of 'dechnology' and 'humart' is used to evaluate how designers incorporate the idea of dechnology, as well as the interwoven experience of humart, in design process, and can also be used for investigating Taiwan's design development [13].

### 2.1. Taiwan's Economic Development

Taiwan's economic development can be represented in a smile curve depicting the three progressing stages of OEM (original equipment manufacturer), ODM (original design manufacturer), and OBM (original brand manufacturer), as show in Figure 1. The three stages also reflect the tendency of Taiwan's economic development, transforming from globalization, localization, to glocalization, respectively [13]. Chien et al. [19] used a framework with five "Fs", function, friendly, fun, fancy, and feeling to demonstrate the transformation from designing "function" for users' need in the early 20th century to accommodating "feeling" for the users' pleasure in the 21st century. In the early days, designers design based on the concept of "form follows function". Today, technology advancement has provided users with complete new forms of experience. The human-centered concept has changed designers' perception from 'design for function' to 'design for feeling'. Modern product design should not merely aim to satisfy the functional needs of consumers, but instead it must also consider the affectional feeling of users.

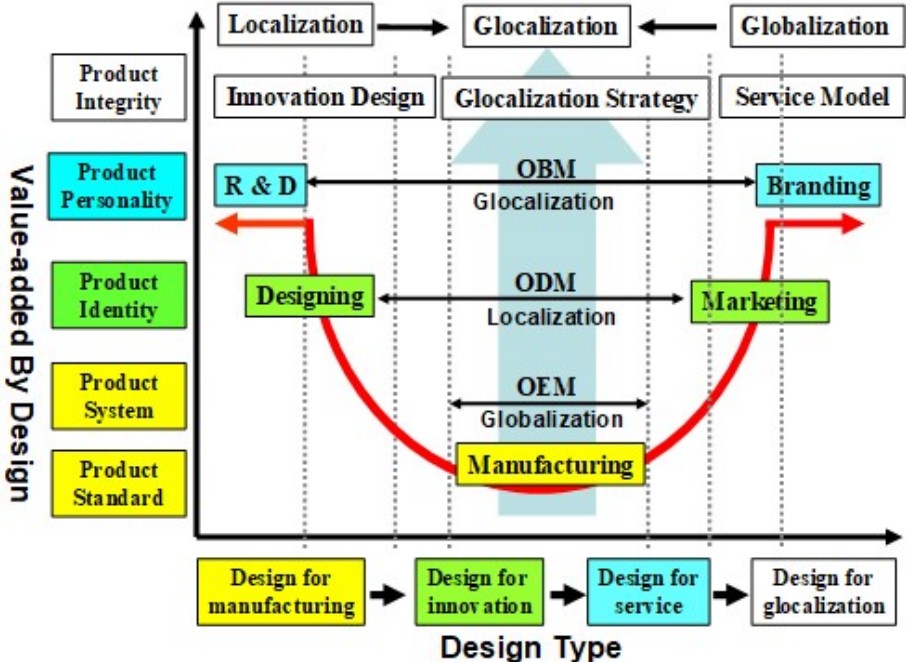

**Figure 1.** Taiwan's economic development. (Reprinted with permission from [13]. Copyright 2015 Lin, R. et al.)

### 2.2. Taiwan's Design Development

Taiwan's design development is similar to the change in design trend [20]. Taiwan's three economic development stages also reflect the trend of its design development as it starts with 'function' to 'feeling' (emotional design), 'use' to 'user' (human-centered design), and ends in 'hi-tech' to 'hi-touch' (user-experience design), a process of evolution that represents the adaptiveness of Taiwan's design development.

During the OEM stage, designers tend to only think of products' functionality when it comes to "design". As the economy moved to ODM stage, technological advancement provided users with completely new delivery system between "design (use)" and "service

(user)". Finally, in OBM stage, "feeling" was incorporated into product design to reflect the emotional aspect of user experiences [20]. Subsequently, "design for feeling" has become the key element for evaluating innovative products.

Taiwan was known globally for its achievements in information technology (IT) industry and the production for "Hi-Tech" 3Cs (computer, communication, and consumer) electronic products and peripherals. With respect to the IT industry's 3Cs, cultural and creative design industries have their own 4Cs (cultural, collective, cheerful, and creative). The 4Cs are the key components for evaluating 'hi-touch' products as cross-cultural factors have become more important for design strategy in the world (Figure 2). The intersection of 'design driven innovation' and 'humanity in arts' is the core value of products. Lin et al. [3] pointed out that in the knowledge economy era, the interconnection between culture and industry is increasingly close.

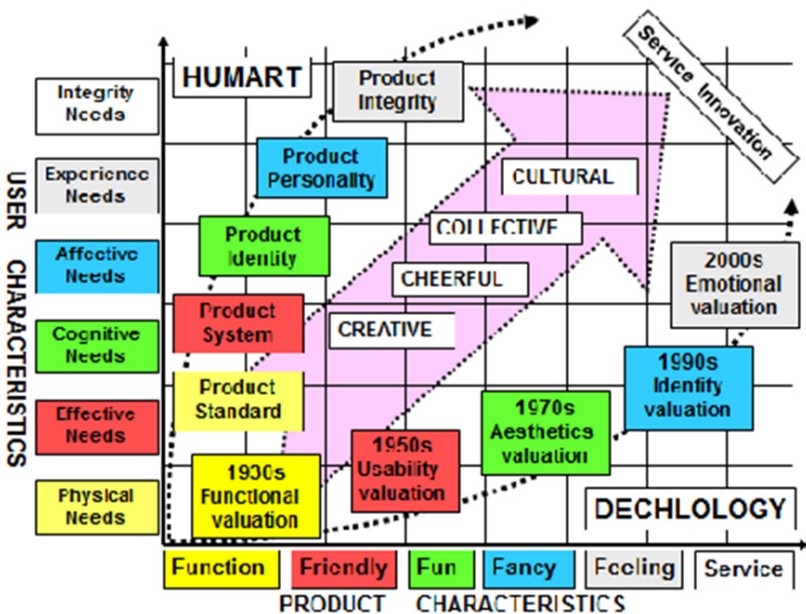

**Figure 2.** Framework of cultural creative industry's 4Cs. (Reprinted with permission from [5]. Copyright 2014 Lin, R. et al.)

## 3. From STEM to STEAM in Taiwan's Secondary School Education

As Taiwan's economy progressed, its educational reformation also underwent major transformation. In the mid-1990s, arts and aesthetic education became popular in the world. The global education system was moving towards creative thinking, cultural diversity, and interdisciplinary. The United States passed National Standards for Arts Education to enhance its people's aesthetic literacy. Japan introduced arts appreciation education to its primary school curriculum to emphasize the importance of aesthetic comprehension and localization. In contrast, Hong Kong's aesthetic education was introduced by the Hong Kong Institute of Aesthetic Education (HKIAE), a non-governmental social welfare organization, in 2001 to its primary and secondary school curriculum. For Taiwan, the Ministry of Education (MOE) issued the White Paper on Art Educational Policy in 2005 demanding secondary schools in TVET and 12-year basic education to include at least 4 credits and 12 credits of arts respectively in the curriculum. Later in 2013, MOE launched Aesthetics Education Program, a ten-year long government initiative, to cultivate students' general literacy in aestheticism, establish aesthetic education support system, and strengthen educators' aesthetic knowledge [21].

Although the Taiwanese government did not issue any official policy on STEM per se, STEM was widely adopted by secondary school educational programs as an important teaching concept in the mid-2010, and it had become a new trend for developing school curriculum [21,22]. The core value of STEM was the integration between hands-on and

minds-on learning, such as problem-based and inquiry-based solving. 'Technology' and 'engineering' in particular not only shaped the economy of the country, it also created the needs for preparing students to learn and develop 21st century skills. However, studies also show that gender inequality was an important issue with STEM education, and women's enrollment in these disciplines is lower than men [6,8]. This growing gap not only hinders the potential for innovation, but also creates a negative impact on the economy.

### 3.1. The Gender Gap in STEM

Globally, women are underrepresented in STEM not only as students, but also as teachers and workers [23]. Increasing the number of women studying and working in STEM fields can achieve better solutions to the global challenges because the lack of female participation greatly reduces the diversity of perspectives and the ability to offer different answers to new problems [6]. However, it is not the level of achievement or competency between girls and boys that leads to this gap; it is gender disparities in interest towards STEM that are substantial. Lee [7] argues that as girls get older, especially when they enter secondary school, they start to lose interest in STEM because of their low self-confidence in their own potential, parents and educators' expectation and influence, and the perception from society on these disciplines. Research from Colorado State University and San Diego State University show that women's low participation in STEM disciplines is caused by their lack of confidence in mathematics [24]. A study from Duke University shows that women often are the minority in STEM-related classes and they experience a sense of loneliness [7]. Craig et al. [25] used the narrative research method to study parents' influence on students' employment in STEM, and the result shows that people who were guided and cultivated by parents at a younger age would access STEM-related job opportunities from multiple paths. At the societal level, social and cultural norms also have a strong impact on the roles of women and men play in society. These gender stereotypes have great impacts on girls' learning in STEM [24].

### 3.2. STEAM, a Solution to Gender Gap

Unlike traditional STEM, STEAM is a new theoretical model for science educators and curriculum developers whose core value is the integration and promotion of creativity with rationalization [26,27]. STEAM can be understood as the integration of STEM with arts, such as design, creative thinking, and any interdisciplinary education approach, and emphasizes the synergy of creativity and problem-solving skills. Although the concept of STEAM involves a more holistic agenda of contributing towards a sustainable development goal as a global citizen, this study only concentrates on its impact on gender equality and industry development. By introducing arts to STEM, the gap of gender equality becomes smaller. Many initiatives were conducted in the world to increase women's interest and confidence in the fields of science and technology. Among them, Girls4STEM was a project carried out in Spain to raise girls' interest from childhood through interacting with female role models which are referred to as "STEM experts" with their family [6]. In Taiwan, an initiative called 'Female Scientist Cultivation' was sponsored by the Ministry of Science and Technology (MIST)'s 'Gender in Science Technology' Project in 2017 to raise the awareness of gender bias and promote the synergy between women and science through playing board game across the nation [28]. Based on Chu [29]'s study, the STEAM approach increases primary and secondary level students' competency in observation, problem identification, and problem solving. If the learning began in elementary school, issues with gender inequality and stereotyped bias could be minimized. Subsequently, the Education Department of New Taipei City launched a weekend activity called 'FUM Coding' to encourage girls in the age 7–15 and their parents to experience coding exercises under the guidance of renowned female computer coders whom were referred to as "hero" in the session. The purpose was to improve interest in learning through idol admiration. The result shows increase in girls' interest in STEAM, as well as the mothers'.

### 3.3. Informal Training as Supplement to STEAM

Out-of-school time activities and informal learning environments (campuses, camps, etc.) have been shown to be important factors that can make a difference to increase students' interest in STEM disciplines [30]. When the concept of STEAM was first introduced to Taiwan's secondary school, the system was faced with various challenges. Educators were mostly unprepared for the change and unfamiliar with knowledge beyond their disciplines. Schools in general lacked the fund and resources needed to carry out the courses [29,31]. Hence, informal training outside of school emerged as a supplement to the transition. Both profit and non-profit educational organizations offered students with after-school online and offline courses on themes like robot, maker movement, and math. Other private entities such as Openlab Taipei and Fablab Taipei, as well as some colleges and high schools, provided spaces equipped with free resources such as open source software, design software, electronic parts, and tools where people, not restricting to students only, could gather to discuss, produce, and realize the concept of learning-by-doing [32]. Museum and gallery's collections are important sources for the teaching of design thinking and aesthetic literacy in STEAM, but most schools, if not all, did not have direct access to them. To help promoting STEAM in education, Taipei National Palace Museum offered secondary level educators various workshops to enhance their knowledge in teaching; exercises range from historical artifacts appreciation, understanding of artifacts' structure and production process from different disciplinary perspectives, to analysis of cultural creative products through hands-on model building. In one of the workshops, participants were each given a replica of Chinese 'Duke Mao Tripod', and were asked to make a potted plant that measures temperature and humidity [33].

### 3.4. STEAM as an Integration of Analytical and Creative Thinking

As global competition increased, the interconnection between local culture and the global market became closer. Cross-cultural factors increased in importance for design strategy in the global economy, hence the argument between STEM and STEAM became a popular subject in both local design and the global market that were worthy of in-depth study. Traditional STEM focuses on convergent thinking skills while STEAM education focuses on divergent skills [34,35]. Convergent and divergent thinking are two poles on a spectrum of cognitive approaches to problems and questions [36]. Convergent thinking is bias to assume that a question has one right answer and that a problem has a single solution. On the other hand, divergent thinking resists the "accepted" ways of doing things and seeks multiple perspectives and multiple possible answers to questions and problems [37]. However, education must not only foster problem-solving skills, but also the skills to better prepare students for both analytical and creative thinking. Often, solutions produced by this type of thinking are unique and surprising. Boy [38] addressed that human-centered design (HCD) can contribute to the improvement of STEAM approach as it not only provides skills to benefit the practical training, but it is also an integrated approach to learning-by-doing for understanding complex systems. Figure 3 shows Taiwan's adaption of STEM and STEAM education system in accordance with its economic and design development. Taiwan used to cultivate the younger generation with STEM, but it presented a problem of gender inequality, which could hinder Taiwan's economic growth. Now, it has moved to STEAM and diverted its focus to creative education [39,40]. As Land [41] once said, design educators should consolidate the major initiatives in STEM and rationalize the values of arts integration, and encourage students to go full STEAM ahead.

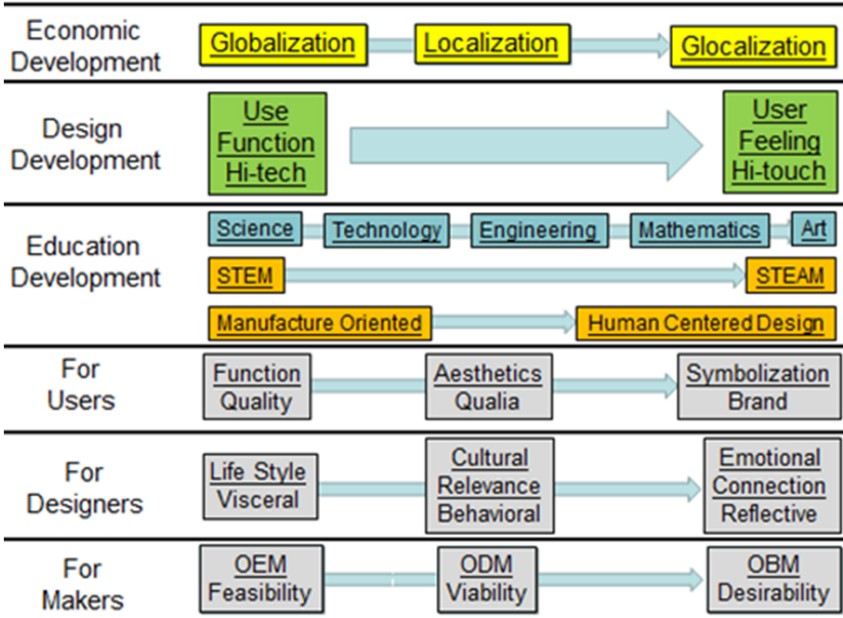

**Figure 3.** Conceptual framework of "from STEM to STEAM" in Taiwan (Source: this study).

## 4. From STEAM to CHEER in Taiwan's Design Education

Although STEM was widely adopted in Taiwan's secondary level education, the issues with low female enrollment buried the potential for future innovation, and the absence of 'arts' in the development of creative thinking refrained artistic skills from integrating into the creativity program. Hence, STEAM was proposed to not only improve gender equality, but also to foster creativity that promotes the approaches to arts subjects. STEAM is a creative skill set of arts which allows the teachers to implement the procedures of enhancing creativity at schools through interdisciplinary approaches [42].

### 4.1. SAD in Higher Design Education

Tsao and Lin [43] conducted a survey at three highly rated design programs in the US and five design programs in Europe, and discovered that the discipline of design education in Taiwan has its long tradition rooted in the field of engineering. Thus, designers in Taiwan were primarily from engineering colleges and their focus studies during secondary school were in STEM. It was rare to have art or social science courses in Taiwan's design education system. The situation is different in the US as Norman [44] described in his core77 column that industrial design is usually established in the schools of art, and taught as a practice with the terminal degree of BA, MA, or MFA. As a result, the skills acquired by Taiwanese designers were not well-suited to meet modern time demand. Consequently, the education system moved to a newer STEAM approach.

In the past, the primary concern of design education was about designers' competency rather than the power of design. Design thinking has become increasingly popular over the years because it is proven to be the right approach not only for the success of design problem-solving, but also for daily activities. Although science and art education are mandatory in Taiwan's 12-year basic education system and children are required to study them starting from primary school, the importance of design is still long neglected.

Based on a Root Cause Analysis (RCA) report, Cross [45] summarized and drew some conclusions on the nature of design. He believes that the focus of design research is on the design process, which can be achieved through an understanding of design cognition, or the 'designerly' ways of knowing and thinking [46]. During the past decades, the aim of design research shifted from creating a 'design science' to that of a 'design discipline' [46]. For example, Shneiderman [47] used three SED elements (science, engineering, and design) to 'Achieving Breakthrough Collaborations' in his book entitled The new ABCs of research:

Achieving Breakthrough Collaborations, in which the ABC of the title stands for 'applied and basic combined', as shown in Figure 4.

**Figure 4.** The new ABCs research framework. (Reprinted with permission from [47]. Copyright 2016 Shneiderman, B.)

Kreifeldt et al. [12] proposed an approach on the basis of Shneiderman's studies of SED [47] to combine science, arts and design (SAD) and built an interdisciplinary concept called 'SAD' for higher design education. As previously mentioned, divergent thinking seeks multiple perspectives and possible answers to questions and problems in design thinking [36]. The best design thinkers must work alongside with other disciplines; many of them have significant experience in more than one discipline, including engineers, marketers, anthropologists, and industrial designers, even architects and psychologists [48]. For 'SAD' to be realized in higher education, design thinking must focus on design problem-solving.

*4.2. CHEER in Design Practice*

Tsao and Lin [43] recommended that arts should be included in design domain in Taiwan's design education. They think that teamwork between designers and other professionals, internship programs for the students, diversifications of students' course work, design as a process of project management, transformation of cultural features into design, and closer collaboration between academia and industry are all important agendas to prepare college students for post-graduate employment. For example, starting from the 1990s, Taiwan government launched more than three five-year plans to upgrade its industrial design competency. In the schemes, working models were established by experienced design scholars and students from universities, and they were later used for setting up working patterns by design students on specific projects based on the enterprises' real needs [49].

CHEER (collaboration, humanity, empathy, ecology, and renaissance) is a concept for cultivating the collaboration between 'humanity' and 'art' in design practice. 'C' represents the interdisciplinary 'collaboration' between dechnology and humart during Taiwan's economic development [13]. 'H' and 'E' represent 'humanity' and 'empathy', which are derived from the concepts of "use to user", "function to feeling" and "hi-tech to hi-touch" stages of Taiwan's design development. The other 'E' means the ecology for exploring the relationship between dechnology and humart when they are merged into design thinking.

Finally, 'R' is the symbolic achievement of reaching the design "renaissance" when the purpose of design is met.

### 4.3. A Framework of STEAM to CHEER

This study proposes a research framework as shown in Figure 5 that may have appeared to be a "new" theory, but in fact, it is the regrouping and restructuring of previously known design concepts in communication. The framework illustrates the interwoven relationship between STEAM and CHEER in Taiwan's design education and provides a guideline to reach the design goal of CHEER. Thinking about design as a process of communication, the framework also explores the relationship between designers and users. Figure 5 shows design education as a matrix of STEAM and CHEER for evaluating design education. Lin et al. [50] examined the ways designers communicate with users in the design process and provide a better understanding of designer–user communication in the social context and the interactive experience.

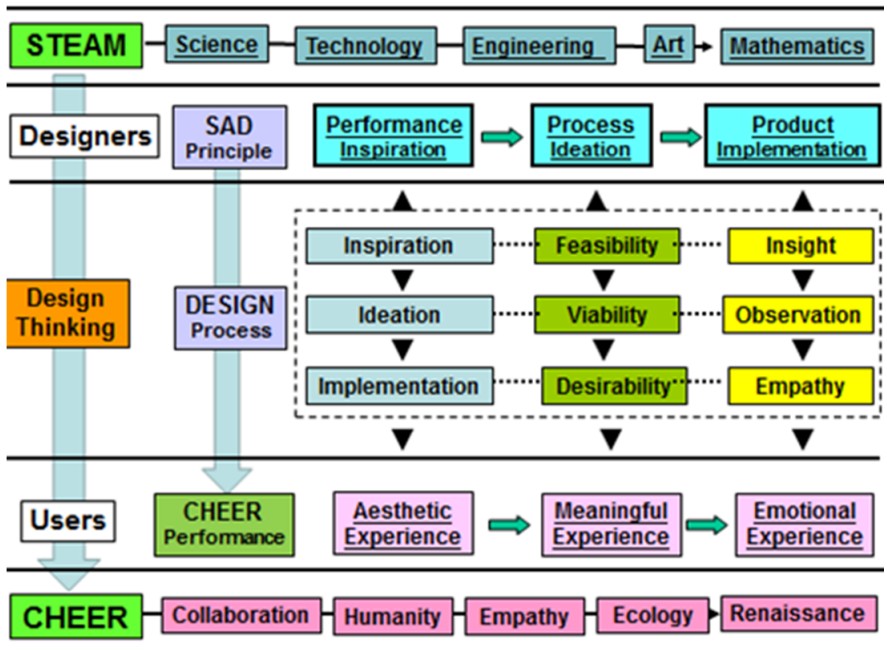

**Figure 5.** A framework of STEAM to CHEER. (Reprinted with permission from [50]. Copyright 2009 Lin, R. et al.)

Starting from the top of the matrix (STEAM), designers are educated based on the principles of SAD, which includes three key stages for designers to implement: performance (inspiration), process (ideation), and product (implementation). "Performance" is the inspiration to produce a significant design. "Process" represents the designers' ideation that can be reproduced. "Product" is the implementation of products' signification and expression which can be delivered to the users [13,43]. Next, design thinking starts with the design process that embodies three perspectives: technical feasibility, economic viability, and human desirability [48]. "Feasibility" is a product's functionality; "viability" indicates what a product can achieve under the company's sustainable business model; and "desirability" represents how much a product appeals to customers. Hence, the core value of CHEER lies in the intersection of technical feasibility, economic viability, and human desirability of the users. For users' 'CHEER', Norman [44] proposes three levels of design process—visceral, behavioral, and reflective levels that correspond to three kinds of user's experience, namely aesthetic, meaningful, and emotional experience respectively, as shown on the bottom of Figure 5 [51]. This users' experience can be used for evaluating product design.

## 5. Some Case Studies for Validating STEAM to CHEER Model

Many studies had explored the successful transformation of Taiwan's economic development in the 1980s. Chen and Bei [52] conducted a case study on Franz Collection Inc., a Taiwan-based porcelain company which had gone through all three economic stages, and explored the managerial and manufacturing challenges it had faced during the establishment of a global brand. Other researchers focused on a variety of fields which include industry policies, R&D, upgrading, and innovation strategies and others [14]. Few studies have looked into Taiwan's design development in greater details. Therefore, a case study was selected to validate the success of Taiwan's design development.

Kreifeldt et al. [12] conducted a program called Design Team In Training (DTIT), which was an internship program designed to bridge the gap between academia and industry, to reach CHEER at the National Taiwan University of Arts (NTUA) (Figure 6). This program was intended to explore the relationship between SAD in higher education and CHEER in practice through case studies. For SAD to be validated in higher education, DTIT must be conducted with real design students in a real working environment. The approach provided a platform to examine the manners in which designers apply and incorporate the principles of SAD and the interwoven experience of CHEER in their design process. For example, a project of designing 'pleasure' into keyboards was carried out through the DTIT program [12]. One of the keyboard designs won the 2006 Japan G-Mark Award and received the remarks "original, appealing, fresh, and creative life experience's design" from the jury. This result presents a paradigm for validating SAD in college design education and CHEER in design practice for future design education. The DTIT program not only allows design students to gain professional experience by participating in business-oriented design projects, the hosting company also has the opportunity to cultivate their employees through enrolling in the program [43].

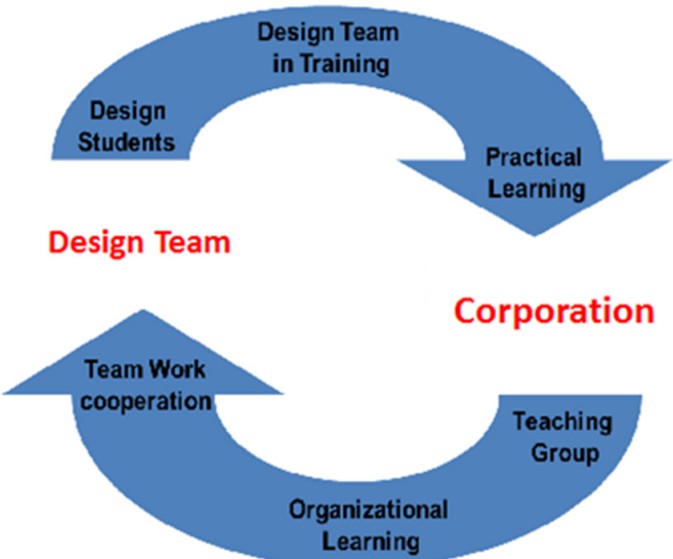

**Figure 6.** A framework of Design Team in Training (DTIT). (Reprinted with permission from [12]. Copyright 2018 Kreifeldt, J. et al.)

## 6. Conclusions and Suggestions

This study first reviews Taiwan's economic and design development and identifies the three stages, OEM (use to user), ODM (function to feeling), and OBM (design for feeling) to illustrate how a developing country transforms from 'local culture' to 'global market' through its design education reform. During the transition, its secondary level education system moved from STEM to STEAM as a result of higher TVET skills demand. Finally, the study validates that by adopting the STEAM approach, the system can reach SAD in higher education, and consequently to CHEER in design practice through case studies.

While Taiwan's economy was rapidly growing, its education system was undergoing major transformation. Taiwan's education system first adopted STEM in the secondary school to produce abundant skilled workforce in response to the country's high labor demand. Yet, like many other developed countries in the world, gender inequality in STEM-related studies and workplace emerged as an issue that could hinder the potential of growth and innovation. Thus, the Taiwanese government took initiatives to include 'arts' into STEM curriculum to increase women's enrollment and promoted creative thinking and process. Eventually, the country's secondary education system moved to a more gender-equal, dynamic, and interdisciplinary STEAM approach, which not only prepares the younger generation with 21st century skills, but also emphasizes the importance of solving community and environmental issues as global-citizens. With the help of formal and informal educational training provided by public and private sectors, Taiwan's education system was able to overcome challenges during the transition. As a result, women's enrollment and interest in STEM-related disciplines increased, and like men they would be prepared with 21st century skills. Together, they created a strong workforce that pushed Taiwan's economy even further and stronger.

As global competition increases, all government are taking necessary measures to ensure their long-term economic growth. Developing countries like the ASEAN have recognized the important role STEM plays in TVET education, and they are eager to find solutions that can effectively increase their peoples' access to quality technical skills and development programs [4]. Vuong [53] suggests that the Vietnamese government should promote the education of science to the general public, and take measures to encourage female participation in tertiary level of education to scale down labor waste. Goy [54] thinks that reducing the attrition rate irrespective of gender in Malaysia's STEM education is a cost-effective measure to fulfil the employment needs in various industries that support the dynamic economy. Indonesian Minister of Research, Technology, and Higher Education Mohamad Nasir encourages the country's 12 public TVET universities to collaborate with Taiwan to enhance its skilled labors' literacy in STEM-related fields. Therefore, the result and implications provided by this study can serve as a reference for other countries in the region who share similar cultural and socioeconomic development and future goals as Taiwan.

**Author Contributions:** Data curation, R.L.; Resources, J.H.; Writing—original draft, C.L. All authors have read and agreed to the published version of the manuscript.

**Funding:** This research received no external funding.

**Institutional Review Board Statement:** Not applicable.

**Informed Consent Statement:** Informed consent was obtained from all subjects involved in the study.

**Data Availability Statement:** The authors confirm that the data supporting the findings of this study are available within the article.

**Conflicts of Interest:** The authors declare no conflict of interest.

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
