# Peer review of "From STEAM to CHEER: A Case Study of Design Education Development in Taiwan"

_education, doi:10.3390/educsci11040171_

Round 1

Reviewer 1 Report

The article presents an interesting review and historical analysis of the transformation of design education regarding STEM education transition to STEAM. The framework could help the international community as Taiwan seems to be a paradigm of a technologically innovative country.

Major change:

However, the authors seem to underestimate the value of STEAM education per se. STEAM education offers students the opportunity to develop 21st century skills as citizens; it is not only an approach to creating more creative expert workers. I propose adding such a notice to make crystal that they have not underestimated the aspect of STEAM, but they choose to concentrate on its impact on the industry.

Minor changes:

88: Correct the reference [45 to [45]

199: “For examples” to “For example” or something similar

250: “…and innovation strategies [42]…etc.” to “…[42] and others”

Reviewer 2 Report

Dear authors,
congratulations on his interesting work.

After reviewing the manuscript, from my point of view, there are still some important points that you can improve to achieve a more powerful work.

I list some aspects that I found:

Abstract:

There is an abuse of the word development and the verb develop. 

I think it is convenient to highlight the main objective of the article and the results and conclusions. What is the presented proposal about: a bibliographic review of the state of the art? What are the implications of your work in education? Does the work focus only on higher education?

General remarks:

Section 1:

What is the objective of the work? What are the implications for education?

Lines 42-49: “Based on Taiwan’s historical development and other previous studies [30, 31, 32, 36, 42 45, 47], this article intends to first explore how Taiwan’s economic development advanced from ‘function’ to ‘feeling’ and consequently inspired its design education development. Later, case studies are applied to substantiate Taiwan’s design development as a fusion of ‘function’ and ‘feeling’ and a process of design evolution [28, 36, 47]. The result shows that the interwoven relationship between ‘globalization’ and ‘localization’ in Taiwan’s design education development are the key factor to the success of Taiwan’s economic development”: 

I think the reader can get confused. It seems that half the work is written and contains references and concepts about economics and design development, but there is a lack of implications regarding society, students, teachers, etc. The article is addressed to an education journal and this is important because of the audience.

Section 2.1, 2.2, and 2.3: They are based exclusively on reference 45. From my point of view, this is a weakness. I think you abuse using figures 1, 2, or 3 (same type)

Section 3:

what level of education are you dealing with?

Are elementary school, high school, and college levels included?

Lines 119-122: “As educational reformation in Taiwan progressed (YEARS, PERIOS OF TIME, WHEN?), STEM (Science, Technology, Engineering, and Mathematics) was widely adopted by educational programs as an important teaching concept, and it had become a new trend for developing schools’ curriculum [1, 5, 20, 37, 52]” . Here you are refeering to elementary schools.

Lines 142-143 “Traditional STEM focuses on convergent thinking skills while STEAM education focuses on divergent skills [39, 50]. Education must not only foster problem-solving skills,”. It is difficult for the reader to follow. You must provide definitions of what convergent and divergent skills are.

STEM education includes problem-solving and inquiry-based methods that are connected to creativity.

When we talk about STEM and/or STEAM methodologies and education, the implications of these methodologies for students are usually discussed taking into account the effect of gender, age and other characteristics. Is there any work on this done in Taiwan? 

With what kinds of initiatives do you go from STEM to STEAM? Are they all formal training? Despite the focus of the work in Taiwan, a little bibliography regarding other countries and a description of the results of experiences in different places is missing. This would give more richness to the paper. For example, some works that talk about STEM education deal with the problem of scientific-technological vocations, especially in the case of girls. This has a lot to do with sustainability and the achievement of sustainable development goals in terms of gender equality and diversity, education, etc.

I miss a balance between the manuscript sections. Maybe section 2 is long and section 3 is too short. 

If the training in the design area is in engineering degrees, there is usually a bias and women do not access this type of study. Can you say anything about it? I think it is interesting that these ideas are contemplated since everything contributes to the economic development of the countries.

The change from STEAM to CHEER, what educational level does it affect? Maybe figure 9 should include this look.

It is important to detail whether art/design education is included in formal or informal training (after-school classes, informal programs, etc.). On many occasions, informal activities make students increase their interest in STEM areas and have more options to study engineering degrees, an aspect that is related to access to design education. On the other hand, this training may be related to sustainability education.

I think these references can help you open new horizons in this work or future work:

  • Formal or informal training of students: It also has an impact on the study of gender equity and the design of quality educational interventions that are aspects that are included in the Sustainable Development Goals (SDGs). I recommend that you read interventions in this regard: https://www.mdpi.com/2071-1050/12/15/6051 (STEM education, informal learning and gender), https://www.mdpi.com/2071-1050/12/10/4283 (citizen science, formal training, SDGs)
  • Gender gap and STEM (https://ieeexplore.ieee.org/document/9137264

Section 4

Lines 149-151: From STEAM to CHEER in Taiwan’s Design Education Although STEM was widely adopted in Taiwan’s design education, the absence of  ‘Arts’ in the development of creative thinking refrained artistic skills from integrating into the creativity program. School? Higher education?  When you talk about design, which subjects an educational levels are you including?

In section 4.1

“Tsao & Lin [59] conducted a survey at three highly rated design programs in the US and five design programs in Europe, and found out that the discipline of design education in Taiwan has its long tradition rooted in the field of engineering.” You are talking about Higher education.

Section 6:

As a conclusion it is said that (lines 289-291) that: “the approach can serve as a great opportunity for Taiwan to market itself in the world and share its experience in design education development with the international community.”, but in the work there has been no recollection of what happens in design in other countries. What would Taiwan's contribution to design education be for other countries?

When do you talk about design, which subjects and educational levels are you including?

Other aspects:

  • Improve the quality of figure 6.
  • Improve conclusions and suggestions. There are only 2 paragraphs: this is not well balanced with respect to the text. I’m not sure if a figure should be included in the conclusions.
  • There are too many references. It’s complicated for the reader to follow the discourse of the paper looking so many times at the references section.
  • Please, the order of references in the text, should be improved. I think you should use the order of appearance in the text.

Round 2

Reviewer 1 Report

Thank you for the revisions. 

Reviewer 2 Report

Dear authors,

you have made a big effort and your work has greatly improved. It is an interesting paper, and I really liked reading it, but I have noticed important details about the citations of works used in the manuscript and the selected bibliographic references.

1) Please, be careful: the sentence "Lack of female professionals also reduces the diversity of perspectives and the ability to offer different answers and breadth to new problems" and for this reason, you should include number reference there (at the end of the sentence).

2) To refer to the idea "women’s performance in math and science is not much different from men", I think you should choose a reference such as "Kersey, A.J.; Braham, E.J.; Csumitta, K.D.; Libertus, M.E.; Cantlon, J.F. No intrinsic gender differences in children’s earliest numerical abilities. NPJ Sci. Learn. 2018, 3, 1–10" https://www.nature.com/articles/s41539-018-0028-7 This is a work published in an important journal that surely has followed a peer review process and is a good reference used in studies, conferences, etc. that deal with topics similar to the idea that you put forward in that sentence. It will serve you for your future jobs.

3) Reference [8] is kind of a conference report and I'm not sure about the peer review process. From my point of view, a good choice of bibliographic references is essential. I suggest you use a reference like  https://ieeexplore.ieee.org/document/9137264 to support the ideas in your manuscript where you use ref [8] or at least, use https://ieeexplore.ieee.org/document/9137264 to better support the arguments.

4) Please, take care when you cite a work done by several authors, you should use the formula "first author et al.". (when you used ref [8], I think you missed "et al.")

These are my main concerns at this stage. I think authors can correct these issues easily.

Kind regards

Round 3

Reviewer 2 Report

Dear authors, 

congratulations. The manuscript has greatly improved and has become a powerful paper. 

Just a few remarks to improve the format about citations in the text and typos in references:

Citations in the text:

Line 176: "Lee [6] argues that as girls get..." ==> reference number 6 or 7?

Some typos in References:

Ref 6: (surnames authors and pages: please note accents and symbols in López-Iñesta and Garcia instead of Carcia)

Benavent, X.; de Ves, E.; Forte, A.; Botella-Mascarell, C.; López-Iñesta, E.; Rueda, S.; Roger, S.; Perez, J.; Portales, C.; Dura E.; Garcia-Costa, D.; Marzal, P. (2020). Girls4STEM: Gender diversity in STEM for a sustainable future. Sustainability12(15), 6051.

Ref 30: (surnames authors)

López-Iñesta, E.; Botella, C.; Rueda, S.; Forte, A.; Marzal, P.

Congratulations, I will follow your work. You almost got it.

Regards
